# Introduction, Dispersal, and Predominance of SARS-CoV-2 Delta Variant in Rio Grande do Sul, Brazil: A Retrospective Analysis

**DOI:** 10.3390/microorganisms11122938

**Published:** 2023-12-07

**Authors:** Thaís Regina y Castro, Bruna C. Piccoli, Andressa A. Vieira, Bruna C. Casarin, Luíza F. Tessele, Richard S. Salvato, Tatiana S. Gregianini, Leticia G. Martins, Paola Cristina Resende, Elisa C. Pereira, Filipe R. R. Moreira, Jaqueline G. de Jesus, Ana Paula Seerig, Marcos Antonio O. Lobato, Marli M. A. de Campos, Juliana S. Goularte, Mariana S. da Silva, Meriane Demoliner, Micheli Filippi, Vyctoria M. A. Góes Pereira, Alexandre V. Schwarzbold, Fernando R. Spilki, Priscila A. Trindade

**Affiliations:** 1Laboratório de Biologia Molecular e Bioinformática Aplicadas a Microbiologia Clínica, Departamento de Análises Clínicas e Toxicológicas, Universidade Federal de Santa Maria, Santa Maria 97105-900, Brazil; 2Centro Estadual de Vigilância em Saúde, Secretaria Estadual da Saúde do Rio Grande do Sul (CEVS/SES-RS), Porto Alegre 90610-000, Brazil; 3Laboratório de Vírus Respiratórios e Sarampo, Instituto Oswaldo Cruz Institute, Fundação Oswaldo Cruz (FIOCRUZ), Rio de Janeiro 21040-360, Brazil; 4Departamento de Genética, Instituto de Biologia, Universidade Federal do Rio de Janeiro, Rio de Janeiro 21941-853, Brazil; 5Instituto de Medicina Tropical, Faculdade de Medicina da Universidade de São Paulo, São Paulo 05508-220, Brazil; 6Vigilância em Saúde, Secretaria Municipal da Saúde de Santa Maria, Santa Maria 97060-001, Brazil; 7Departamento de Saúde Coletiva, Universidade Federal de Santa Maria, Santa Maria 97105-900, Brazil; 8Departamento de Análises Clínicas e Toxicológicas, Universidade Federal de Santa Maria, Santa Maria 97105-900, Brazil; 9Laboratório de Microbiologia Molecular, Universidade FEEVALE, Novo Hamburgo 93510-235, Brazil; 10Departamento de Clínica Médica, Universidade Federal de Santa Maria, Santa Maria 97105-900, Brazil

**Keywords:** genomic surveillance, viral lineages, variants of concern, genome analysis, phylogeny

## Abstract

Mutations in the SARS-CoV-2 genome can alter the virus’ fitness, leading to the emergence of variants of concern (VOC). In Brazil, the Gamma variant dominated the pandemic in the first half of 2021, and from June onwards, the first cases of Delta infection were documented. Here, we investigate the introduction and dispersal of the Delta variant in the RS state by sequencing 1077 SARS-CoV-2-positive samples from June to October 2021. Of these samples, 34.7% were identified as Gamma and 65.3% as Delta. Notably, 99.2% of Delta sequences were clustered within the 21J lineage, forming a significant Brazilian clade. The estimated clock rate was 5.97 × 10^−4^ substitutions per site per year. The Delta variant was first reported on 17 June in the Vinhedos Basalto microregion and rapidly spread, accounting for over 70% of cases within nine weeks. Despite this, the number of cases and deaths remained stable, possibly due to vaccination, prior infections, and the continued mandatory mask use. In conclusion, our study provides insights into the Delta variant circulating in the RS state, highlighting the importance of genomic surveillance for monitoring viral evolution, even when the impact of new variants may be less severe in a given region.

## 1. Introduction

At the end of January 2021, new cases of COVID-19 emerged in various districts of Maharashtra, India, after a four-month hiatus. Subsequently, India experienced a second wave of the COVID-19 pandemic, resulting in more than 400,000 deaths. This surge was attributed to the emergence of lineage B.1.617 [1], primarily sublineages B.1.617.1 (Kappa variant) and B.1.617.2 (Delta variant). The Delta variant swiftly spread worldwide [2].

In May 2021, the World Health Organization (WHO) designated the Delta variant as a VOC [3]. Additionally, four other SARS-CoV-2 VOCs have been identified: Alpha [4], Beta [5], Gamma [6], and Omicron [7]. Due to global travel and transit, VOCs have been both imported into and exported from their countries of origin, circulating worldwide. Delta was detected in Brazil in May 2021 by a ship crew of Asian origin that had docked in the state of Maranhão [8]. By July 2021, autochthonous transmission of this variant was evidenced in Rio de Janeiro [9]. This marked the first of several independent importation events involving Delta. By September 2021, Delta had become the dominant lineage in the pandemic landscape across the southeast, northeast, and south regions of Brazil [10].

The first case of SARS-CoV-2 infection in Rio Grande do Sul (RS) was reported on 10 March 2020. RS, located in the southernmost region of Brazil, occupies 3% of the nation’s total land area. It ranks as the ninth-largest state in Brazil and shares its borders with two neighboring countries, Uruguay and Argentina [11]. Among its 497 cities, 196 are situated along the border strip, including 11 twin cities [12]. RS is the fifth most populous state in the country, trailing behind São Paulo, Minas Gerais, Rio de Janeiro, and Bahia [13]. The state’s economy is based primarily on agriculture, livestock, and industry, positioning RS as the fourth-largest economic contributor in Brazil [14]. Moreover, RS has notable social indicators, including low infant mortality rates and high life expectancy and literacy rates, all of which contribute to its status as one of the states with the highest quality of life in Brazil [15]. Up until the study period, the state had confirmed over 1,436,734 cases of SARS-CoV-2 infection, with an incidence rate of 12,725.72 per 100,000 inhabitants and more than 35,203 deaths attributed to complications of the COVID-19 [16].

Mutations and genetic recombination events within the SARS-CoV-2 genome, particularly in the spike protein (S), have the potential to impact the virus’ fitness. These events may result in increased transmissibility, evasion of the host’s immune response, and reduced effectiveness of certain vaccines and treatments [17]. The Delta variant, in particular, harbors a set of mutations in the S protein, including L452R, T478K, and P681R, all of which have contributed to the increased transmissibility of this variant [18]. Notably, the L452R mutation within the S receptor-binding motif (RBM) [19] has been shown to enhance infectivity while reducing sensitivity to neutralizing responses [1]. These cumulative mutations have raised the S protein’s affinity for angiotensin-converting enzyme 2 (ACE2), resulting in decreased vaccine efficacy compared to the Alpha and Beta variants that were prevalent before the emergence of Delta [18]. As the Delta variant continued to evolve, it gave rise to subvariants, such as AY.1, characterized by an additional mutation in the S protein, K417N. This mutation confers heightened antibody escape properties, increased transmissibility, and a strong affinity for lung epithelial cells [20]. Subsequently, numerous Delta subvariants have emerged, potentially carrying altered biological characteristics and clinical manifestations [21].

Detection and genomic surveillance of infectious diseases, such as COVID-19, are crucial for responding to the pandemic. Next-generation sequencing (NGS) technologies, which enable timely whole-genome sequencing (WGS), have become one of the most commonly used tools in virus research. They serve as a common approach for pathogen identification and tracking, establishing transmission routes, monitoring virus evolution, and controlling outbreaks [22,23]. In Brazil, less than 1% of SARS-CoV-2-positive cases are sequenced [24]. Furthermore, the vast territorial expanse of the country and the decentralized health surveillance system pose challenges in monitoring, expanding, and connecting regional genomic data. In this study, we conducted an extensive and robust analysis of the introduction and dispersal of the Delta variant in the RS state. To achieve this, we sequenced 1077 SARS-CoV-2-positive samples collected from June to October 2021 and analyzed the whole-genome sequences with regard to phylogenetic profiles and polymorphisms, as well as the dispersion of the variant across the state.

## 2. Materials and Methods

### 2.1. Study Population

The study population consisted of SARS-CoV-2-positive patients in the state of RS, Brazil, from 13 June to 9 October 2021, spanning epidemiological weeks (EW) 24 to 40. The study received approval from the Research Ethics Committee of the Universidade Federal de Santa Maria (Certificado de Apresentação de Apreciação Ética—CAAE: 51019821.0.0000.5346) and Universidade FEEVALE (Certificado de Apresentação de Apreciação Ética—CAAE: 33202820.7.1001.5348).

### 2.2. Epidemiological and Clinical Data

Epidemiological data for SARS-CoV-2-positive patients were obtained from DATASUS, in accordance with patient data protection laws. We assessed data related to age, gender, city, and the date of the SARS-CoV-2 positive test. Additionally, the number of confirmed cases and deaths and vaccination data were retrieved from public government databases [16,25].

### 2.3. Sample Screening and Genome Sequencing

SARS-CoV-2-positive samples from public and private laboratories in the RS state were sent to the Laboratório de Biologia Molecular e Bioinformática Aplicada a Microbiologia Clínica (LABIOMIC), Universidade Federal de Santa Maria, Santa Maria (*n* = 260); the Laboratório de Microbiologia Molecular (LMM), Universidade FEEVALE, Novo Hamburgo (*n* = 263); or Fundação Oswaldo Cruz (FIOCRUZ), Rio de Janeiro (*n* = 554), Brazil. Samples with a cycle threshold (Ct) ≤ 30 were pre-selected for sequencing. Due to the high number of SARS-CoV-2-positive patients, the pre-selected samples were randomly chosen, taking into account the sequencing capacity of the laboratories. Library construction was performed as described in the ARTIC protocol version 3 [26], the QIAseq SARS-CoV-2 Primer Panel (QIAGEN, Hilden, Germany) for paired library enrichment and QIAseq FX DNA Library UDI Kit (QIAGEN, Hilden, Germany), or the Illumina COVIDSeq Test (Illumina, San Diego, CA, USA). Genomes were sequenced using the MinION device (Oxford Nanopore Technologies, Oxford, UK) or Illumina MiSeq (Illumina, San Diego, CA, USA). MinKNOW v.21.05.12 was used to collect raw data from the MinION, which underwent high-accuracy base-calling and quality control analyses using Guppy v5.0.12. Illumina raw data were processed in Basespace (https://basespace.illumina.com, accessed on 17 October 2022).

#### 2.3.1. Viral Genome Assembly

The assembly of the consensus sequence was conducted using the nCoV-2019 novel coronavirus bioinformatics protocol [27] along with Minimap2 version 2.8.3 [28] and BCFtools version 1.7-2 [29], Geneious Prime™ version 2022.2 [30], or ViralFlow 0.0.6 [31]. To compare the sequences in subsequent analyses, those with over 75% genomic coverage, fewer than 3000 undetermined nucleotides (N), and those rated as either good or mediocre by Nextclade were selected. Sequences failing to meet these standards were excluded from our analysis.

#### 2.3.2. Lineage Identification

The clade and the number of gap regions in the viral consensus sequences were analyzed using Nextclade version 2.4.1 [32]. The lineages of the viral consensus sequences were also determined using Pangolin version 4.1.1 and pangolin-data version 1.13 [33].

#### 2.3.3. Single Nucleotide Polymorphism Identification

To identify SNPs, insertions, and deletions, Geneious Prime™ [30] was employed. In brief, the FASTA-assembled consensus sequences were imported and aligned with the reference genome (NC_045512.2). The ‘Find Variations/SNPs’ tool was configured with a minimum variant frequency of 0.25.

### 2.4. Spatio-Temporal Analysis

For the spatio-temporal analysis, we searched for sequences that had location indications in metadata deposited in GISAID [34,35] between 13 June and 9 October 2021. However, only the samples sequenced in this study met this criterion. The coordinates of the 30 health microregions (Verdes Campos, Entre Rios, Fronteira Oeste, Belas Praias, Bons Ventos, Paranhana, Vale dos Sinos, Vale do Caí, Carbonífera, Capital/Vale do Gravataí, Sete Povos das Missões, Portal das Missões, Região da Diversidade, Fronteira Noroeste, Caminho das Águas, Alto Uruguai Gaúcho, Região do Planalto, Região das Araucárias, Região do Botucaraí, Rota da Produção, Região Sul, Pampa, Caxias, Campos de Cima da Serra, Vinhedos, Uvas e Vales, Jacuí/Centro, Santa Cruz do Sul, Vale das Montanhas, and Vale da Luz) [36] (Appendix A) were combined with the results of the genotyped samples, and a model of the Gamma variant replacement and Delta variant dispersal in the RS state was built. Kernel density analysis was performed using QGIS v3.26 software [37]. The cartographic base was provided by the Instituto Brasileiro de Geografia e Estatística (IBGE) [38].

### 2.5. Phylogenetic Analysis

#### 2.5.1. Data Sets

Two data sets were compiled in accordance with Fonseca et al. [39]. Both data sets encompassed all Delta variant sequences (Pango lineage AY.*, NextStrain clade 21J) captured on 20 October 2022 (*n* = 5006) from GISAID. For the proportional target data set, samples from all Brazilian states were randomly selected in proportion to the estimated number of Delta variant cases per week in each state during EW 24 to 40. To achieve this, all high-quality complete genome sequences from Brazil between 13 June and 9 October 2021 were downloaded and categorized by variant and state. Using this categorization, we calculated the frequency of Delta infection cases by dividing the total number of SARS-CoV-2 sequences by the number of Delta variant sequences per week in each state. Estimates of Delta variant infection cases were determined by multiplying the frequency by the corresponding total number of confirmed cases per week in each state, as provided by the government [40]. The estimated numbers of Delta variant cases per week in each state were then divided by the total estimated cases of this variant and multiplied by 5000 (approximately the number of NextStrain clade 21J sequences) to obtain the number of target samples. In cases where a state had more Delta variant sequences than the target number, we generated a selection of random numbers using the random() function in Python. Sequences were manually inspected using AliView v.1.28 [41], and duplicate sequences were removed. Conversely, when the target number exceeded the number of Delta variant sequences available, we included all of them (detailed calculations are available in Appendix A). This selection resulted in 5008 samples, including 1077 from RS, totaling 10,014 sequences. For the uniform targets data set, we used the median of weekly Delta variant sequences for RS over the study period. This corresponded to selecting 40 targets per week in each state, either randomly or by including all available sequences, as in the previous data set. This selection resulted in 7025 samples, including 1077 from RS, totaling 12,031 sequences.

#### 2.5.2. Analysis of Temporal Signal and Identification of Brazilian Clades

Each data set, along with the Wuhan-Hu reference genome (EPI_ISL_402124), was aligned using Pangolin with default settings. A maximum-likelihood phylogenetic construction was conducted using IQ-Tree v2.1.2 [42] under the GTR+F+I+G4 nucleotide substitution model [43,44] and a Shimodaira–Hasegawa-like approximate likelihood ratio (SH-aLRT) branch test [45]. The constructed tree was manually rooted with the oldest sequence in FigTree [46]. The maximum-likelihood tree was examined in TempEst v1.5.3 [47] to identify samples with inconsistent temporal signals, i.e., in the root-to-tip regression, sequences that deviated more than 1.5 times the interquartile range of the residual distribution were considered outliers. The outliers were removed, and the rooted tree was evaluated with the TreeTime [48] mugration method using a discrete asymmetric two-state model (Brazil/International). Figure 1 provides a comprehensive diagram of the study methodology.

## 3. Results

### 3.1. COVID-19 Overview

We analyzed the dispersal of the Delta variant in the RS state between EW 24 and 40 (June to October 2021). During this period, the RS state government registered 164,194 new confirmed cases of SARS-CoV-2 infection. There was a mean of 7440 ± 5143 confirmed cases and 181 ± 188 deaths (Figure 2A). Despite the introduction of the Delta variant in the state, the data do not indicate an increase in confirmed cases or deaths from COVID-19.

The introduction of the previous VOC was accompanied by mandatory mask use and large-scale population SARS-CoV-2 vaccination (Figure 2B). It is estimated that on 13 June 2021, 6,305,296 vaccines were administered, and by 9 October, this number reached 15,033,655. These numbers represent 39.17% and 15.82% of the total RS population vaccinated with the first and second doses, respectively (EW 24). In addition, 0.0009% received a single dose. Starting from EW 27, the percentage of individuals vaccinated with the third dose increased significantly, and from EW 37 onwards, the number of those who received the additional dose (previously vaccinated with the single dose) also increased. Moreover, during the evaluated period, 19 individuals received a fourth dose.

### 3.2. Epidemiological Aspects of Delta Introduction Period

Among the SARS-CoV-2-positive samples, a total of 1077 were sequenced to delineate the introduction, dispersal, and prevalence of the Delta variant. The mean genome coverage achieved in our analysis was 98.1%, encompassing a range from 75.1 to 100%, relative to the 29,903 bp of the NC_045512 reference genome. The average depth of our sequencing was 1159×, with a variability spectrum extending from 20 to 8019 (Appendix A). Within this sample set, 422 genomes were classified as Gamma and 648 genomes were designated as Delta (Appendix A for further details). Additionally, our analysis identified one case of Alpha, two cases of Lambda, as well as one case each of B and B.1 variants. Patients infected with the Gamma variant exhibited a median age of 43, whereas those infected with the Delta variant had a slightly lower median age of 41, as detailed in Table 1. The majority of these patients were adults, comprising 76.8% for the Gamma variant and 66.7% for the Delta variant, with a female predominance.

### 3.3. SARS-CoV-2 Delta Introduction and Dispersal

The first case of infection with the Delta variant was pinpointed on 17 June, during EW 24 (Figure 3). The Gamma variant exhibited a prominent presence, constituting more than 70% of the reported cases in the RS state during the period spanning EW 24 to 28. Nevertheless, there was a gradual increase in cases attributed to the Delta variant. By the time EW 32 arrived, a total of 26 cases of the Gamma variant were documented, accounting for 41.9% of the overall caseload. Into EW 33, a shift in the viral landscape became evident, marked by a transition in the predominant variants. The Delta variant established itself as the predominant variant, representing more than 70% of all SARS-CoV-2-positive cases.

When we detailed the genetic composition of the 1077 sequenced samples, twenty-four Pangolin lineages were identified (Figure 4). The most frequently detected lineage was AY.99.2 (440 sequences, 40.85%), followed by P.1 (356 sequences, 33.05%), AY.101 (162 sequences, 15.04%), P.1.2 (35 sequences, 3.25%), AY.43.2 (22 sequences, 2.04%), P.1.7 (13 sequences, 1.21%), and AY.100 (12 sequences, 1.11%) (Figure 4B). The remaining ten lineages had a frequency lower than 1%, collectively representing less than 3.5% of the sequences.

### 3.4. Single Nucleotide Polymorphism

The genome-wide mutation profile of the Gamma and Delta variants sequenced in the RS state is depicted in Figure 5. The Gamma variant exhibited 38 mutations, comprising 30 single nucleotide polymorphisms (SNPs—18 transitions and 12 transversions), 2 substitutions, 1 insertion, and 5 deletions (Appendix A). Of these, 5.3% (2) were located in the 5′-UTR, 36.8% (14) in the ORF1ab, 31.6% (12) in the S gene, 2.6% (1) in ORF3a, 2.6% (1) in ORF8, 7.9% (3) in the N gene, 2.6% (1) in the intergenic UTR, and 10.5% (4) in the 3’-UTR (Figure 5A and Appendix A). The Delta variant presented a higher number of mutations, amounting to 45, including 42 SNPs (32 transitions and 10 transversions) and 3 deletions (Appendix A). Among these, 4.4% (2) were found in the 5’-UTR, 48.9% (22) in the ORF1ab, 17.8% (8) in the S gene, 4.42% (2) in ORF3a, 2.2% (1) in the M gene, 6.7% (3) in ORF7a, 2.2% (1) in ORF7b, 2.2% (1) in ORF8, 8.9% (4) in the N gene, and 2.2% (1) in the intergenic UTR (Figure 5B and Appendix A). Notably, 74.2% (23) and 73.8% (31) of the SNPs occurring in the Gamma and Delta variant coding regions, respectively, were non-synonymous, particularly those occurring in the S gene (Appendix A). The Gamma variant circulating in RS did not possess the S84L mutation (ORF8) characteristic of this variant (Appendix A). The Delta variant circulating in RS did not exhibit the E156G (S gene), S84L (ORF8), and T60A (ORF9b) mutations. However, it did have the additional mutations I1091V (ORF1a), T4087I (ORF1a), and A23V (ORF3a).

### 3.5. Phylogenetic Analysis

To compare the SARS-CoV-2 strains circulating in RS with those in the rest of Brazil and the world, we conducted phylogenetic analyses. Firstly, we conducted a maximum likelihood phylogenetic analysis using the proportional data set (Figure 6A). Within this analysis, 63.9% of the Delta sequences clustered in a large Brazilian clade, encompassing a total of 3025 sequences. The remaining sequences were distributed throughout the tree. Subsequently, we isolated the large Brazilian clade and generated a new maximum likelihood tree. This tree exhibited a significant temporal signal (R^2^ = 0.21; β = 9.09 × 10^−4^) (Figure 6B). A Bayesian analysis, utilizing 10% of these sequences, estimated a rate of 5.43 × 10^−4^ substitutions/site/year (95 per cent HPD median: 4.41 × 10^−4^–6.42 × 10^−4^) (Figure 6C). These analyses were replicated with a uniform data set, yielding similar results (Appendix A).

We assessed the retraction of the Gamma variant and the dispersal of the Delta variant within the state of RS by analyzing the locations of infected patients’ residences (Figure 7). The pandemic landscape was predominantly characterized by the Gamma variant. Within the EWs under scrutiny in this study, EW 26 recorded the highest percentage of territories with the presence of the Gamma variant, accounting for 63.3%. On average, Gamma variant samples were identified across 25.88% of the state. A turning point occurred on 17 June during EW 24 when the first case of Delta variant infection was documented in a patient originating from the city of Garibaldi, within the Vinhedos e Basalto Microregion (Figure 7B). Interestingly, the subsequent week (EW 25) showed no instances of Delta variant infection reported. Starting from the onset of EW 26, the incidence of Delta variant infections experienced a steady ascent, permeating throughout the entirety of RS. In EW 27, an additional 13 cases of Delta variant infections were identified, spanning multiple health microregions, including Capital/Vale do Gravataí, Caxias, Fronteira Oeste, Região da Diversidade, Região do Planalto, and Vale do Caí (Figure 7B). The majority of Delta cases was observed in the Capital/Vale do Gravataí region (23.64%), closely followed by Alto Uruguai Gaúcho (17.57%), with Vale dos Sinos (11.51%) and Vale do Caí (11.2%) also reporting significant numbers (Figure 7B). By EW 39, the Delta variant had reached a maximum presence of 63.3% across the state’s territory, with an average of 26.3% coverage. These data highlight the presence of gaps in the state’s surveillance system as all municipalities in the region reported individuals infected with SARS-CoV-2, yet the specific variant and/or subvariant remained unidentified. Predominantly, the south and west regions accounted for the most uncovered in this regard.

## 4. Discussion

In this study, we conducted a retrospective analysis using a large data set to identify the introduction and dispersal of the Delta variant in the RS state. The previously circulating Gamma variant led to approximately a 4-fold increase in the number of SARS-CoV-2-positive cases and approximately a 6.7-fold increase in deaths [16]. However, the replacement of Gamma with Delta did not result in an alteration in the incidence of cases and deaths. This stability likely resulted from natural and/or vaccinated immunity, particularly in the RS population [25,49]. When the Delta variant was introduced in the RS state on 17 June, 39.16% of the total population had received the first dose of the vaccine. The RS state had already experienced COVID-19 waves, leading a significant portion of the population to acquire natural immunity against SARS-CoV-2. Of particular note, the use of masks became mandatory on 20 March 2020 [50] and remained in effect during the period of the introduction and spread of the Delta variant. This situation differed from that of many European and Asian countries facing the third wave of SARS-CoV-2 (previous VOC Delta) [51,52,53]. The Brazilian government faces challenges in the domain of Public Health Surveillance. During our study period, we documented eight individuals who had received a fourth vaccine dose, even though the official campaign only began in December 2021. 

The Delta variant rapidly dispersed, comprising 71% of the SARS-CoV-2-positive cases by EW 33 (15–21 August 2021), nine weeks after the first case. This finding aligns with a study involving 183 SARS-CoV-2-positive samples, which reported that the RS state reached this milestone in August [49]. Based on sequences accessible through GISAID, Mayer et al. have further substantiated these findings [54]. Utilizing phylogeographic analysis, it has been determined that the RS state imported the AY.99.1 and AY.99.2 sublineages from Brazil’s southeast region, as well as the AY.101 sublineage from Paraná, which is also part of the southern region of the country [55]. The state of Rio de Janeiro, located in Brazil’s southeast region, reported the first autochthonous transmission of the Delta variant in July. By August, it had become the dominant variant in the state [56]. Minas Gerais, a state bordering Rio de Janeiro, reported 73% of Delta variant infections in September, specifically during EW 37 [39]. In October, the Tocantins [57] and Rondônia [58] states in the northern region of Brazil also reached the same percentage mark. Interestingly, the state of Pará, situated along the border with Tocantins, registered infection with the Gamma variant up until July 2022 [59]. To our knowledge, it was only these five Brazilian states that investigated the replacement of Gamma by the Delta variant. Comprehensive studies analyzing data from across Brazil have established that by September 70% of SARS-CoV-2 positive cases were of the Delta variant [60,61]. 

The Delta variant exhibits greater transmissibility and lower sensitivity to neutralizing antibodies derived from vaccination or prior infection [17]. Our sequences revealed the presence of 90.6% of the distinctive mutations associated with the Delta variant, along with two additional mutations, specifically I1091V and T4087I [49]. These mutations were identified within the ORF1ab region of the Delta variant, which was introduced into Brazil via the state of Rio de Janeiro [9]. The T4087I mutation involves the substitution of a hydrophilic amino acid, threonine, with a hydrophobic amino acid, isoleucine. This change could exert an influence on viral replication and infection rates [62]. Mutations in the spike protein have a significant influence on immune evasion and infectivity. The T19R mutation is associated with a reduction in the efficacy of monoclonal antibodies [63]. The G142D mutation is linked to a higher viral load [64]. The L452R mutation, in combination with T478K, stabilizes the RBD-ACE2 complex, enhancing infectivity, improving the ability to evade the host’s immune response, and increasing transmissibility. The D614G mutation, present in various SARS-CoV-2 variants, is associated with increased virulence and enhances the cleavage rate at the S1/S2 site [65,66,67,68,69,70,71,72,73]. Finally, the P681R mutation, located at the furin cleavage site S1/S2, augments the cleavage rate and cell invasion, thereby increasing infectivity and transmissibility [65,74,75,76]. It is worth noting that there is a lack of consensus among the Covariants, Outbreak.info, and WHO databases regarding defining mutations of this variant. Consequently, it was classified as a non-characteristic mutation, meaning it was not present in any of the databases.

The majority of the sequenced Delta variants in our study were clustered within the 21J lineage, while three variants were clustered in 21I. Employing a Bayesian analysis, we determined a substitution rate of 5.43 × 10^−4^ substitutions per site per year, which translates to approximately 16.24 substitutions per year. Notably, by utilizing a comprehensive data set, we mitigated sampling bias in comparison to the study conducted by Gularte et al., which reported a higher substitution rate of 34.5 substitutions per year due to their smaller sample size of 183 sequences [49]. Ferrareze et al. estimated a substitution rate per site for the AY.99/AY.99.1, AY.99.2, and AY.101 sublineages circulating in RS, revealing rates ranging from 4.0612 to 6.6080 × 10^−4^, which align closely with the values obtained in the current study [55].

The first case of Delta variant infection was documented on 17 June (EW 24) in a patient from Garibaldi city, within the Vinhedos e Basalto health microregion. This finding diverges from the typical pattern of introducing new variants, which often happens in capital cities, metropolitan areas, or cities with major airports [77]. In RS, the introduction of the Delta variant occurred within the Vinhedos e Basalto region, a tourist destination, primarily during the winter season. We postulate that the progress in vaccination and the partial relaxation of COVID-19 prevention and control measures in Brazil facilitated tourism, ultimately leading to the introduction and dispersal of the Delta variant in this region, spanning from EW 24 to 33 of 2021. It is important to highlight the shortcomings in genomic surveillance and the reporting of SARS-CoV-2 infection cases. The COVID-19 pandemic presented an unprecedented challenge for the Brazilian healthcare system and likely for healthcare systems worldwide. Whole-genome sequencing was not readily available to the majority of institutions, and while there has been an increase in the number of institutions adopting this technique, it still falls short of enabling effective genomic surveillance. The absence of coordination and standardized criteria for sample sequencing, resulting in some health microregions being underrepresented or not represented at all. This limitation is evident in our study, where we encountered a scarcity of samples from the west and south regions of the state, hampering our ability to accurately identify the circulating variants in those areas. Despite the absence of data from various regions, our study compiled samples from 140 municipalities within the state, surpassing the previously recorded figures of 44 [55] and 20 municipalities [49].

Although the Delta variant is no longer in active circulation, it remains essential to retrospectively analyze the progression of the COVID-19 pandemic. Such analysis is crucial for the scientific community and governmental authorities to understand the behavior of an emerging virus, identify the most effective control and prevention strategies to mitigate transmission, discern the factors that contributed to the reduction in hospitalizations and deaths, and recognize which approaches were ineffective. This information is vital for understanding how emerging pathogens may behave within the state and for guiding the government in the development of more targeted control and prevention measures for the future.

## 5. Conclusions

Our study provides insights into the introduction and dispersal of the Delta variant in the RS state of Brazil. We observed the replacement of the previously dominant Gamma variant by Delta, which exhibited greater transmissibility and reduced sensitivity to neutralizing antibodies. Despite this shift, the incidence of COVID-19 cases and deaths remained stable, likely due to a combination of natural and vaccinated immunity, as well as the continued mandatory mask use. Sequencing data revealed the presence of characteristic Delta variant mutations, along with additional ones. However, the lack of consensus among different databases regarding defining characteristic mutations underscores the complexity of variant characterization. Phylogenetic analysis and substitution rate estimation provide valuable insights into the viral evolution of the Delta variant in the RS state, addressing previous sampling biases. In summary, our study underscores the need for ongoing genomic surveillance and the importance of maintaining public health measures in the face of emerging SARS-CoV-2 variants. Monitoring and understanding the dynamics of these variants are crucial for effective pandemic response and control.

## Figures and Tables

**Figure 1 microorganisms-11-02938-f001:**
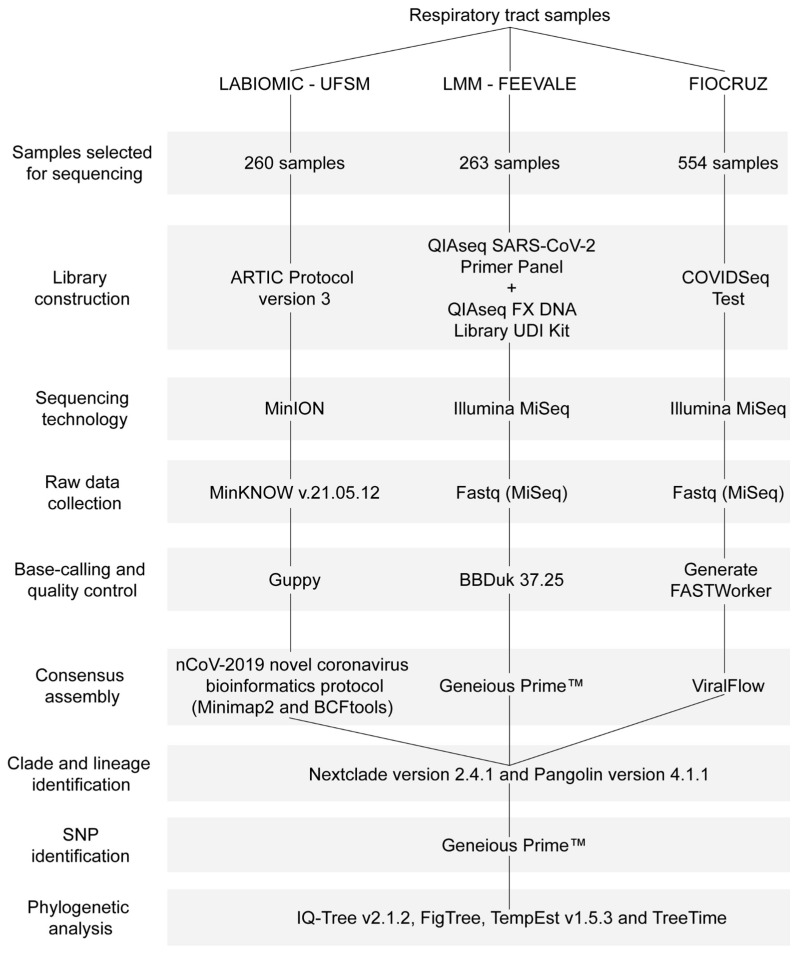
Diagram of the study methodology.

**Figure 2 microorganisms-11-02938-f002:**
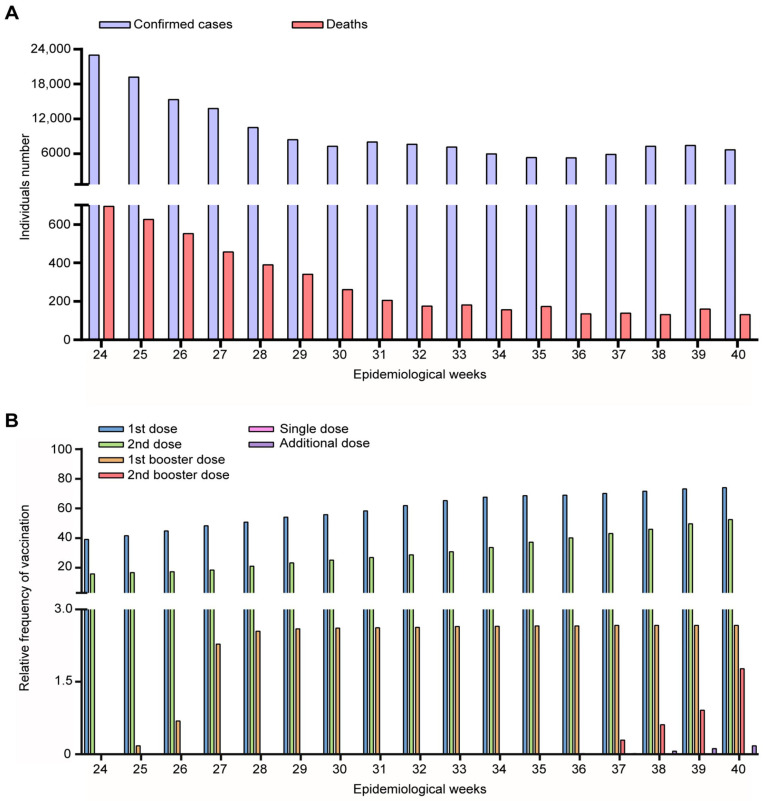
COVID-19 overview in the Rio Grande do Sul state, Brazil, during epidemiological weeks 26 to 40 of 2021. (**A**) number of confirmed cases, deaths, and (**B**) vaccinated individuals.

**Figure 3 microorganisms-11-02938-f003:**
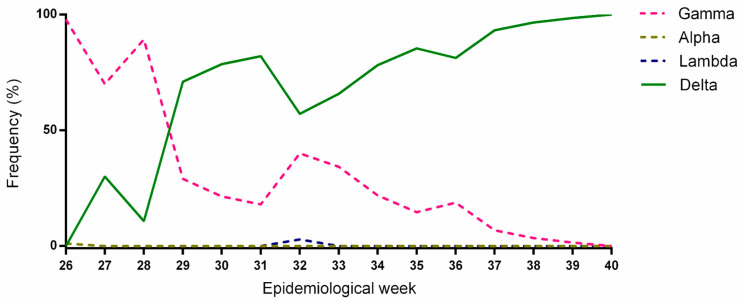
Replacement of Gamma variant by Delta variant considering the sequenced samples in this study.

**Figure 4 microorganisms-11-02938-f004:**
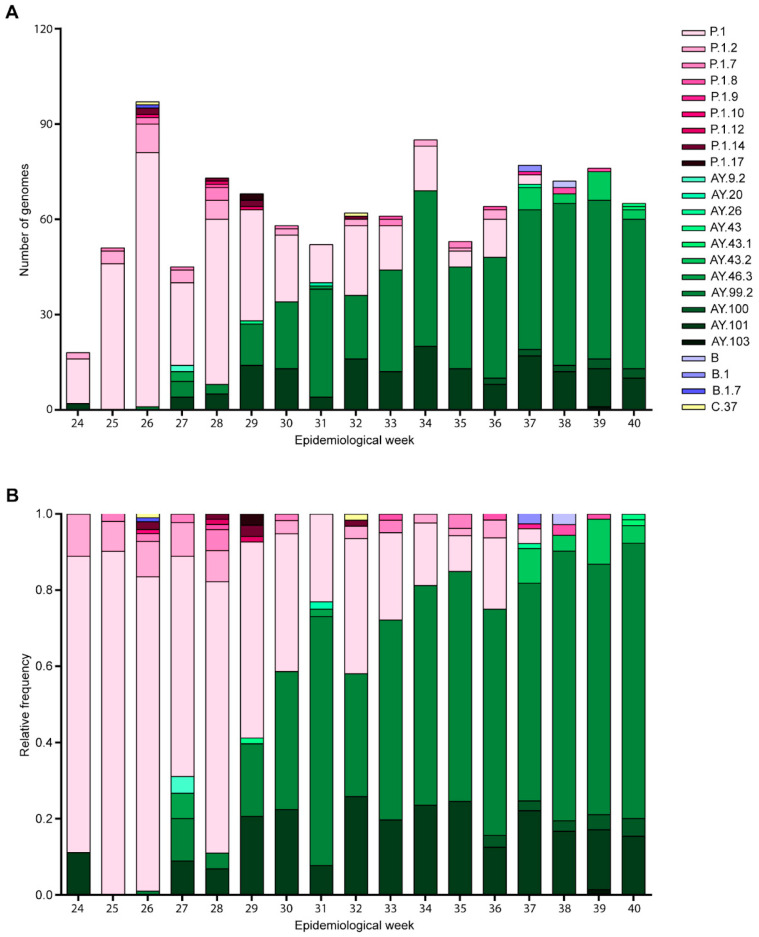
Genetic composition of SARS-CoV-2 lineages in Rio Grande do Sul from June to October 2021. (**A**) absolute variation number in lineages’ frequencies across epidemiological weeks, as classified by Pangolin version 4.1.1; (**B**) lineages’ variation frequencies in relative terms.

**Figure 5 microorganisms-11-02938-f005:**
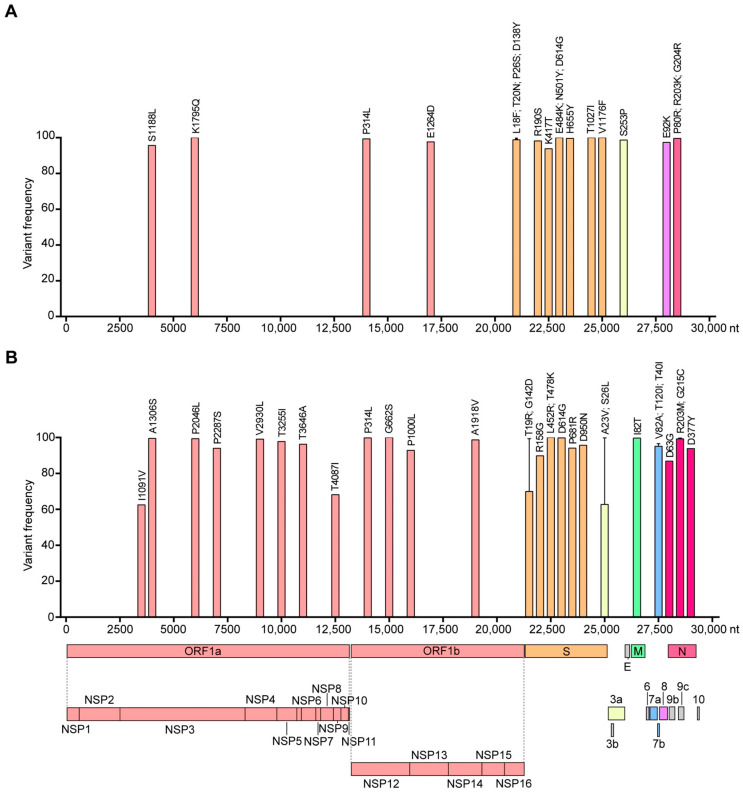
Mutational profile and variant frequency in the coding region of SARS-CoV-2 Gamma (**A**) and Delta (**B**) variants.

**Figure 6 microorganisms-11-02938-f006:**
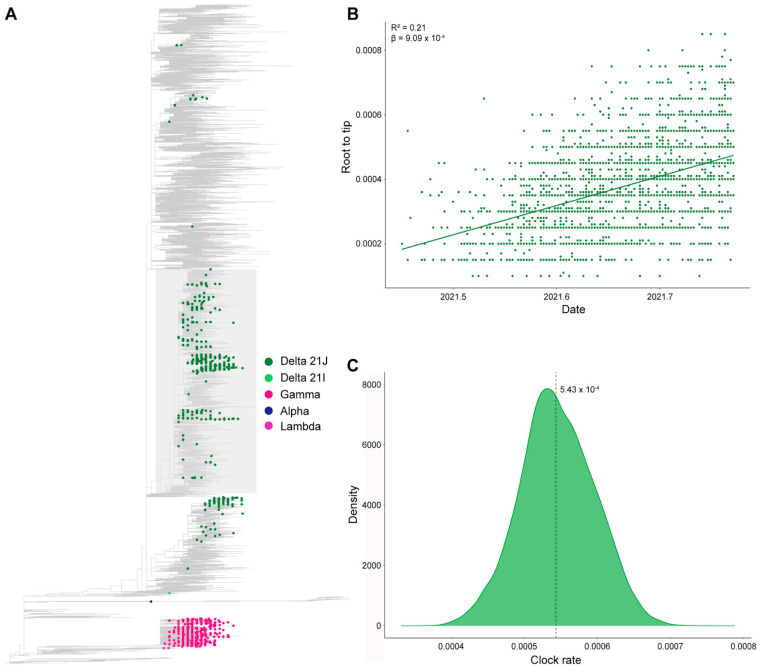
Maximum likelihood phylogeny constructed with proportional data set (*n* = 10,013). In this study, 63.9% of the Delta cases sequenced clustered into a large Brazilian clade (highlighted in gray), covering 3025 sequences (**A**). Temporal signal analysis of the large Brazilian clade through root-to-tip regression (**B**). Clock rate distribution using 10% of the large Brazilian clade sequences (**C**).

**Figure 7 microorganisms-11-02938-f007:**
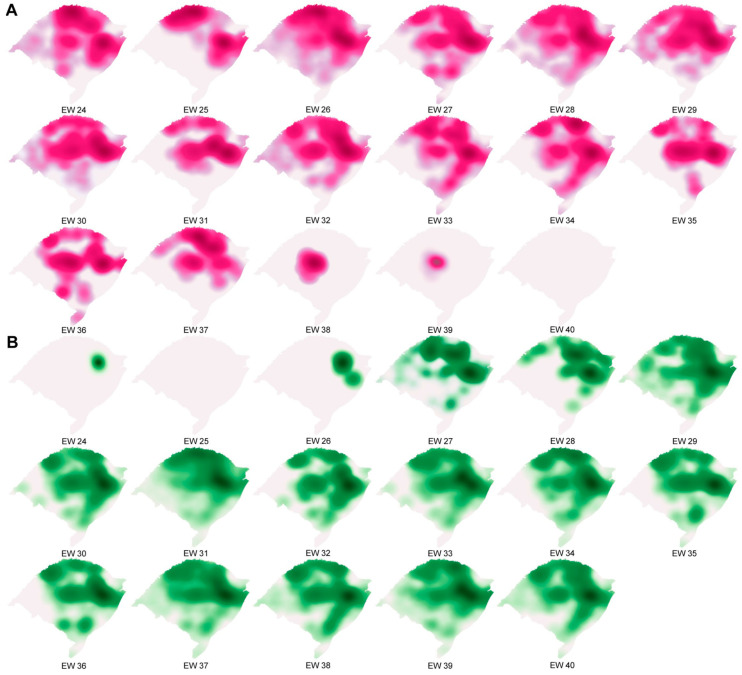
SARS-CoV-2 genomic surveillance in Rio Grande do Sul, Brazil, from June to October 2021. Gamma retraction (**A**) and Delta variant dispersal (**B**) across the state of Rio Grande do Sul considering the H = health microregions in each epidemiological week sampled in our study.

**Table 1 microorganisms-11-02938-t001:** Epidemiological data of SARS-CoV-2-positive patients infected with SARS-CoV-2.

	Gamma Positive (*n* = 422)	Delta Positive (*n* = 648)	Others (*n* = 7)
Age (median)	43	41	37
<18 (number)	17	48	0
18–60	324	432	5
>60	81	167	2
Female (%)	237 (56%)	340 (52%)	5 (71%)

## Data Availability

All SARS-CoV-2 genomic sequences used in this study are available on GISAID (https://gisaid.org/). The list of epicodes is available in Appendix A.

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
