# Peer review of "Introduction, Dispersal, and Predominance of SARS-CoV-2 Delta Variant in Rio Grande do Sul, Brazil: A Retrospective Analysis"

_microorganisms, 2023, doi:10.3390/microorganisms11122938_

Round 1
Reviewer 1 Report
Comments and Suggestions for Authors
I thank the authors for an interesting study! There are several methodological and clarifying questions:
In Section "2.3 of the study, Sample screening and genome sequencing", the authors cite various sequencing protocols, but do not specify how many reads there were at least for each sample (QIAGEN or Illumina). Because of this, it is unclear how different data from different laboratories can be correlated.
I would recommend the authors to add a diagram (design) of the study itself for clarity and to show the scale of the work.
Figure 4. It makes sense to supplement the scheme of key proteins encoded in the genome (superimposed on the genetic coordinates).
Reviewer 2 Report
Comments and Suggestions for Authors
In the study, the authors conducted a comprehensive and robust analysis of the introduction and spread of the VOC delta in the RS state by sequencing a large number of SARS-CoV-2 positive samples collected from June to October and analyzing the whole genome sequences for phylogenetic profiles and polymorphisms. This work is very useful for people to study the spread and predominance of SARS-CoV-2 delta variant, especially in Rio Grande do Sul. There are several concerns need to be addressed before publication.
1. Figure 1 and Figure 3 should be embedded in the right paragraph.
2. D614G is a crucial mutation in S protein of B.1.1 variant which is variant different from VOC delta (B.1.617.2).
3. Please provide the meaning of x-axis in Figure 4
4. Please provide the latest data related to the references 2 and 3.
Reviewer 3 Report
Comments and Suggestions for Authors
Summary:
The paper investigates the introduction and spread of the Delta variant of SARS-CoV-2 in the state of Rio Grande do Sul (RS), Brazil. The study sequenced 1,077 SARS-CoV-2-positive samples collected from June to October 2021 and identified the prevalence of Gamma and Delta variants. The paper notably finds that despite the rapid spread of the Delta variant, the number of cases and deaths remained stable, suggesting the effectiveness of vaccines, prior infections, and mask mandates. Considering the current dominant virus is the lineage or sublineage of the Omicron variant, the paper has its vital limitations.
Paper Strengths
1. Innovative Methodology: The paper employs genome sequencing on a large set of samples to understand the spread of the Delta variant.
2. Empirical Findings: The paper provides a granular view of how the Delta variant has been spreading in RS state, Brazil, which has broader implications for understanding the dynamics of the SARS-CoV-2 virus.
3. Theoretical Insights: The study calculates the estimated clock rate for the mutations, adding to the understanding of viral evolution.
4. Relevance: Given the global impact of COVID-19, the paper’s focus on understanding the dynamics of a variant of concern is highly relevant.
5. Comprehensive Analysis: The paper not only traces the variant but also correlates it with public health metrics like case numbers and death rates.
Paper Weaknesses
1. Limited Geographic Scope: The paper focuses only on RS state, Brazil, which might limit the generalizability of the findings.
2. Inadequate Discussion on Baselines: Lack of comparisons with other known baselines or methods in the field of genomic surveillance.
3. Spike protein analysis: The discussion of Spike protein is not enough. The authors should analyze the mutations on the Spike protein more.
4. Current Relevance: Considering the current dominant virus is the lineage or sublineage of the Omicron variant, the paper has its limitations.
